# Analysis of the Genome of the Heavy Metal Resistant and Hydrocarbon-Degrading Rhizospheric *Pseudomonas qingdaonensis* ZCR6 Strain and Assessment of Its Plant-Growth-Promoting Traits

**DOI:** 10.3390/ijms23010214

**Published:** 2021-12-25

**Authors:** Daria Chlebek, Tomasz Płociniczak, Sara Gobetti, Agata Kumor, Katarzyna Hupert-Kocurek, Magdalena Pacwa-Płociniczak

**Affiliations:** Institute of Biology, Biotechnology and Environmental Protection, Faculty of Natural Sciences, University of Silesia in Katowice, Jagiellońska 28, 40-032 Katowice, Poland; dachlebek@us.edu.pl (D.C.); tomasz.plociniczak@us.edu.pl (T.P.); sara.gobetti@edu.unito.it (S.G.); agata.kumor@us.edu.pl (A.K.); katarzyna.hupert-kocurek@us.edu.pl (K.H.-K.)

**Keywords:** rhizospheric bacteria, *Pseudomonas qingdaonensis*, plant growth promotion, hydrocarbons, heavy metals, biosurfactants, genome mining

## Abstract

The *Pseudomonas qingdaonensis* ZCR6 strain, isolated from the rhizosphere of *Zea mays* growing in soil co-contaminated with hydrocarbons and heavy metals, was investigated for its plant growth promotion, hydrocarbon degradation, and heavy metal resistance. In vitro bioassays confirmed all of the abovementioned properties. ZCR6 was able to produce indole acetic acid (IAA), siderophores, and ammonia, solubilized Ca_3_(PO_4_)_2_, and showed surface active properties and activity of cellulase and very high activity of 1-aminocyclopropane-1-carboxylic acid deaminase (297 nmol α-ketobutyrate mg^−1^ h^−1^). The strain degraded petroleum hydrocarbons (76.52% of the initial hydrocarbon content was degraded) and was resistant to Cd, Zn, and Cu (minimal inhibitory concentrations reached 5, 15, and 10 mM metal, respectively). The genome of the ZCR6 strain consisted of 5,507,067 bp, and a total of 5055 genes were annotated, of which 4943 were protein-coding sequences. Annotation revealed the presence of genes associated with nitrogen fixation, phosphate solubilization, sulfur metabolism, siderophore biosynthesis and uptake, synthesis of IAA, ethylene modulation, heavy metal resistance, exopolysaccharide biosynthesis, and organic compound degradation. Complete characteristics of the ZCR6 strain showed its potential multiway properties for enhancing the phytoremediation of co-contaminated soils. To our knowledge, this is the first analysis of the biotechnological potential of the species *P. qingdaonensis*.

## 1. Introduction

Due to the widespread contamination of the environment with petroleum hydrocarbons and heavy metals posing a critical threat to human health and the environment, finding innovative ways to remove these pollutants has become a priority in the remediation field. One of the most promising strategies is phytoremediation assisted by rhizobacteria and endophytes that are known to promote plant growth and development through various mechanisms, such as the synthesis of phytohormones, lowering the level of ethylene, facilitating the uptake of minerals, increasing the bioavailability of pollutants through the synthesis of surfactants, and, most importantly, reducing the toxicity of organic contaminants through their enzymatic mineralization [1,2,3,4,5]. It was suggested that cooperation between plant-growth-promoting bacteria and hydrocarbon-/heavy-metal-tolerant plants may result in more effective phytoremediation of soil co-contaminated with these compounds [6].

Literature data indicate the clear potential of *Pseudomonas* strains to participate in restorative phytoremediation of polluted areas, which will enhance sustainable agricultural production and environmental protection [7,8]. Among plant-associated bacteria, *Pseudomonas* strains are capable of metabolizing a wide range of hydrocarbons. For example, *Pseudomonas* sp. Ph6 tested by Sun et al. [9] was able to degrade phenanthrene with 81.1% efficiency, whereas the *Pseudomonas* sp. P3 strain used by Zhu et al. [10] showed high potential to degrade naphthalene, phenanthrene, fluorene, pyrene, and benzo(a)pyrene. Several works have also demonstrated the ability of *Pseudomonas* members to degrade crude oil [11]. The positive effect of these microorganisms on plants exposed to stressful conditions is determined by an improvement in their viability and an increase in biomass. The inoculation of two species of willow (*Salix purpurea* and *Salix discolor*) and perennial ryegrass (*Lolium perenne*) with *Pseudomonas putida* PD1 promoted root and shoot growth and protected the plants against the phytotoxic effect of phenanthrene. Furthermore, 25 to 40% removal of xenobiotics from the soil was observed [12]. In turn, the application of *Pseudomonas putida* VM1450 for the clean-up of soil polluted with 2,4-dichlorophenoxyacetic acid resulted in the increased removal of this herbicide from soil and its reduced accumulation in plant tissues [13]. Bakaeva et al. [14] demonstrated that treating barley seeds with strains of the *Pseudomonas* genus increased germination and root growth in soil contaminated with petroleum oil. Apart from the capability of organic xenobiotic utilization and plant growth promotion, some rhizospheric *Pseudomonas* show resistance to heavy metals and possess features that can alter metal bioavailability, which makes them a useful tool in the remediation of heavy-metal-polluted environments [15,16]. These bacteria contribute to the mobilization and uptake of heavy metals by plants and improve plant growth under heavy metal exposure. For example, Płociniczak et al. [17] noted an increase in Zn and Cd accumulation in the shoot tissues of *Silene vulgaris* by 43.8% and 112.6%, respectively, in plants inoculated with *Pseudomonas helmanticensis* H16 compared to control plants treated with bacterial necromass. It is worth emphasizing that *Pseudomonas* species can adapt to various environmental conditions due to abundant genetic diversity [18]. However, knowledge about genome plasticity remains limited. The application of high-throughput sequencing techniques and functional genomics provides a new direction to achieve a deeper understanding of the genetic basis for versatile metabolic potential and environmental adaptation [19].

In this study, a new *Pseudomonas qingdaonensis* ZCR6 strain isolated from the rhizosphere of *Zea mays* growing in soil co-contaminated with hydrocarbons and heavy metals was investigated, and its plant-growth-promoting traits along with its ability to degrade hydrocarbons and resist heavy metals were investigated.

To the best of our knowledge, there is no previous report on rhizosphere-associated *P. qingdaonensis* with the characteristics of hydrocarbon degradation, heavy metal resistance, biosurfactant production, and plant growth promotion.

## 2. Results and Discussion

### 2.1. Isolation and Identification of the Isolate

The isolation and selection of plant-growth-promoting, hydrocarbon-degrading, and heavy-metal-resistant bacteria from the rhizosphere of plants seems to be particularly important for exploiting the bioremediation potential of plant-bacteria systems [11]. In this study, bacterial strain ZCR6 was isolated as one of 26 strains from the rhizosphere of *Zea mays* growing in soil co-contaminated with hydrocarbons and heavy metals in the vicinity of the Coke Plant “Jadwiga” in Zabrze (Upper Silesia, Poland). Among the isolates, the ZCR6 strain showed the highest ACC deaminase activity, very high surface-active and emulsification activity, demonstrated an ability to produce IAA, siderophores, ammonia, and cellulase, and solubilized Ca_3_(PO_4_)_2_ (Table 1). Based on these results, the ZCR6 strain, as the most promising strain, was chosen for further characterization. Phylogenetic analyses showed that the ZCR6 strain is closely related to *Pseudomonas qingdaonensis* JJ3^T^, an aflatoxin-degrading bacterium isolated from peanut rhizospheric soil, the taxonomic name of which is not validated yet (Figure 1) [20]. According to the EZBioCloud database, the closest relatives of strain ZCR6 (OK597188), on the basis of the 16S rRNA gene sequence, were *Pseudomonas qingdaonensis* JJ3^T^ (100%, MG589917), *Pseudomonas brassicae* MAFF 212427^T^ (99.86%, LC514379), and *Pseudomonas defluvii* WCHP16^T^ (99.29%, KY979145) (Appendix A). In the MLSA, the sequence length of three concatenated genes after alignment was 2563 bp. Through the analysis of concatenated 16S rRNA and two housekeeping genes, ZCR6 was placed into the *P. putida* subgroup [21]. For the ZCR6 strain, the highest average nucleotide identity was observed for *P. qingdaonensis* (99.95%), *P. brassicae* (97.30%), and *P. laurentiana* (95.95%) (Figure 1). The genomic comparison using the TrueBac ID tool showed 99.18% similarity between the genomes of the ZCR6 and *P. qingdaonensis* JJ3^T^ strains.

To our knowledge, little is known about the *P. qingdaonensis* strains associated with plants in the context of their use in environmental biotechnology. Therefore, the presented research will help to expand our knowledge about this species, recently isolated for the first time by Wang et al. [20].

### 2.2. Biochemical Characteristics and Cellular Fatty Acid Composition of the ZCR6 Strain

The biochemical characteristics revealed that *P. qingdaonensis* ZCR6 was positive for oxidase, catalase, and arginine dihydrolase, and was able to produce pyoverdine. In turn, it was negative for dehydrogenase, urease, nitrate reductase, and gelatinase, and was unable to hydrolyze starch and produce pyocyanin. The obtained results were consistent with those obtained in [20] for the *P. qingdaonensis* strain JJ3^T^, except for its starch hydrolysis ability and the production of pyoverdine; however, these may be due to natural variability between strains belonging to the same species.

The cellular fatty acid composition of the ZCR6 strain showed a high percentage of C16:0 (30.98%), C16:1ω7c/C16:1ω6c (26.35%), C18:1ω7c/C18:1ω6c (15.44%), and C17:0 cyclo (10.22%) fatty acids. Additionally, the hydroxylated fatty acids C12:0 2OH_,_ C12:0 3OH, and C10:0 3OH were detected, and their abundance was in the range of 3.34–4.63%. Moreover, C12:0 and C19:0 cyclo ω8c fatty acids were present; however, their abundance was low, 2.22% and < 1%, respectively (Table 2). These results differed from the profile obtained for the JJ3^T^ strain, which was dominated by C17:0 cyclo (24%) and C16:0 (21.4%), while C16:1ω7c/C16:1ω6c and C18:1ω7c/C18:1ω6c fatty acids constituted only 11.5% and 10.5% of the total fatty acids, respectively [20]. However, Wang et al. [20] used a three-day-old culture instead of a 24-h-old culture, as is required and recommended in the MIDI-MIS method. To compare our results with JJ3^T^ strain fatty acid methyl esters (FAMEs), analysis of three-day-old ZCR6 culture was performed. The obtained results showed a high similarity of both profiles.

### 2.3. General Features of the Genome

Genome analysis was performed to characterize the genetic potential of this bacterium.

To the best of our knowledge, this is the first study performing a genome analysis of *P. qingdaonensis* species isolated from the rhizosphere of maize. The genome of *P. qingdaonensis* ZCR6 was assembled into 52 contigs consisting of 5,507,067 bp with an average G + C content of 64.5%. Previously, similar genome sizes of different *Pseudomonas* species isolated from the rhizosphere of various plants were reported [20,22,23]. A total of 5055 genes were annotated in the genome, of which 4943 were protein-coding sequences (CDSs). Furthermore, 8 rRNA-, 64 tRNA-, and 4 ncRNA-encoding genes were predicted. A large number of rRNAs is a typical characteristic of microorganisms, allowing them to respond rapidly to changing nutrient availability [24]. In addition, two intact prophage regions of 21.7 kb and 5.9 kb with GC contents of 62.29% and 60.54%, respectively, were identified. It is worth emphasizing that the presence of a prophage in the bacterial genome may increase its resistance to environmental stresses and biofilm formation and support horizontal gene transfer [25]. The general genome features of the ZCR6 strain are described in Table 3. The circular genome representing the entire genome assembly is shown in Figure 2.

The COG database was used to functionally categorize predicted proteins. A total of 4877 protein-encoding genes in the ZCR6 strain were assigned to COG in 21 functional categories (Figure 3). This analysis revealed the highest gene abundance for amino acid transport and metabolism (E, 9.47%), transcription (K, 9.37%), inorganic ion transport (P, 8.35%), signal transduction mechanisms (T, 6.62%), energy production and conversion (C, 6.48%), and cell wall/membrane/envelope biogenesis (M, 6.15%). It is worth noting that the high percentage of genes belonging to the E and P classes may suggest the innate ability of the ZCR6 strain to compete with other microorganisms. It is also worth noting that the high metabolic activity of the studied strain may be reflected in the occurrence of many genes involved in the transcription process [25,26]. Genes belonging to the E and K categories constitute the largest group of genes for other well-known plant growth promotors, *P. fluorescens* PICF7 and *Pseudomonas* sp. UW4 [27,28]. The functional annotation performed using KEGG identified 2945 associated genes (Figure 3). The largest fraction of the predicted genes was assigned to the metabolism of amino acids (327 genes, 11.1%) and carbohydrates (282 genes, 9.57%), as well as to environmental and information processing: membrane transport (146 genes, 4.96%) and signal transduction (140 genes, 4.75%) (Figure 4A). The annotated genes were also assigned and grouped into subsystem feature categories. A clear advantage in the percentage of genes was noted for metabolism cofactors and vitamins (305 genes, 10.4%), amino acids and derivatives (298 genes, 10.2%), protein synthesis (179 genes, 6.14%), and membrane transport (174 genes, 5.9%) (Figure 4B). These results confirmed a preference consistent with the previous results from COG and KEGG functional analysis.

### 2.4. Plant Growth Promotion Potential

#### 2.4.1. Nutrient (N, P, S, Fe) Acquisition

Rhizospheric bacteria use various mechanisms of action to stimulate plant growth. First, due to their direct vicinity of plant roots, they enhance the bioavailability and uptake of beneficial nutrients, such nitrogen (N), phosphorus (P), and sulfur (S). N is an essential nutrient for plant growth, development, and reproduction; however, plants are unable to utilize atmospheric nitrogen (N_2_). Fortunately, some rhizospheric bacteria, called diazotrophs, are able to convert N_2_ to ammonia (NH_3_) in a process called biological nitrogen fixation (BNF) using a complex system known as nitrogenase [5]. Among diazotrophs, the *Azotobacter, Azospirillum, Rhizobium*, and *Bradyrhizobium* genera are the best known [29]; however, species of *Pseudomonas* (e.g., *Pseudomonas stutzeri*) have also been reported [30,31]. The genome of the ZCR6 strain lacks a full set of genes necessary for nitrogen fixation. Nevertheless, the presence of *iscS* and *iscU* genes encoding cysteine desulfurase and nitrogen fixation protein, respectively, with a primary sequence similar to those of NifS and NifU proteins, was reported (Table 4). The role of NifS and NifU is the formation of iron–sulfur (Fe–S) clusters required for the maturation of nitrogenase in nitrogen-fixing microorganisms [32]. Because it was found that functional cross talk between the Nif and ISC systems is possible [33], NifS- and NifU-like proteins from the ZCR6 strain may have been engaged in the formation of Fe–S clusters by other nitrogen-fixing rhizobacteria when the loss of neither NifS nor NifU function occurred. The ZCR6 strain was not able to fix atmospheric nitrogen but showed the ability to produce ammonia (probably via ACCD activity) (Table 1). This mechanism of PGP may also support plant growth by facilitating better supplementation of plants with assimilable forms of N.

Phosphorus is another essential nutrient for plants; however, a large proportion exists in an insoluble form and is thus unavailable for plants. Some soil bacteria play important roles in regulating P availability. This process is mainly mediated by enzymes involved in (1) inorganic P solubilization and organic P mineralization, (2) P uptake and transport, and (3) P starvation response regulation encoded by three corresponding groups of genes [34]. Genome analysis of the ZCR6 strain revealed the presence of genes from all three groups.

The first group contains the *gcd* gene coding for the quinoprotein glucose dehydrogenase (PQQGDH) and the *pqqABCDE* gene cluster necessary for pyrroloquinoline quinone (PQQ) cofactor biosynthesis (Table 4). PQQGDH is an enzyme that is responsible for the biosynthesis of gluconic acid (GA) in the direct oxidation pathway of glucose. The synthesis and secretion of GA is the major mechanism for the solubilization of mineral phosphate [26]. For the ZCR6 strain, P solubilization activity was also confirmed as a halo zone (6.5 mm diameter) on NBRIP agar plates containing an insoluble source of P (Table 1). Additionally, in the genome of ZCR6, the presence of the *ppa* and *ppx-gppA* genes, encoding inorganic pyrophosphatase and exopolyphosphatase, respectively, was detected. Inorganic pyrophosphatase catalyzes the hydrolysis of inorganic pyrophosphate PPi to inorganic phosphate Pi [35], while exopolyphosphatase catalyzes the hydrolysis of the terminal residues of inorganic polyphosphates (polyPs) to Pi [36].

The uptake of phosphate by bacteria occurs via two systems: the low-affinity, constitutively expressed Pit system and the high-affinity, phosphate-starvation-induced Pst system [37]. In the genome of ZCR6, the presence of genes coding for both P uptake and transport systems (*pit* and *pstSCAB*) was reported (Table 4). These systems enable bacteria to assimilate inorganic P under P-rich and P-low conditions [22,38] and have also been described for many environmental strains, e.g., *Pseudomonas psychrotolerans* CS51, *Rhodococcus erythropolis* PR4, *Nocardioides sp.* JS614, *Catenulispora acidiphila* ID 139908^T^, and *Frankia* sp. ACN14a [22,39,40,41,42].

The third group of genes, including *phoB*, *phoR*, and *phoU**,* detected in the genome of ZCR6, was involved in P starvation regulation (Table 4). These genes are components of the Pho regulon, a regulatory circuit involved in bacterial phosphate management. PhoR-PhoB is a two-component regulatory system responsible for detecting and responding to changes in the P concentration in the environment, e.g., when phosphate becomes limited [43]. In this case, PhoR is stimulated to autophosphorylate and then transfer phosphoryl groups to PhoB. Phosphorylated PhoB undergoes a conformational change, allowing it to bind conserved DNA sequences called *pho* boxes in certain promoters, leading to increased transcription of genes, many of which help cells cope with decreased extracellular phosphate levels, e.g., genes involved in P uptake and transport (*pst*). In turn, the precise function of PhoU remains uncertain, and it is speculated that it may function as a negative regulator of the Pho regulon [44].

Sulfur, an essential macroelement required for plant growth, development, and response to environmental changes, is increasingly becoming limiting to plants [45]. Over 95% of the sulfur in soil is organically bound largely in one of two major forms—sulfate esters and sulfonates. These organic forms are not directly available to plants, which rely upon microbes in the rhizosphere for organo-S mobilization [46]. In the genome of the ZCR6 strain, the presence of genes encoding enzymes that are responsible for splitting the O-S bond in sulfate esters (arylsulfatase) and involved in the mobilization of SO_4_ from sulfonates (FMNH_2_-dependent monooxygenase enzyme complex, *ssu* gene cluster) was observed (Table 4). The metabolism of sulfur in the ZCR6 strain also involves the assimilation of inorganic sulfate.

Genome analysis revealed the presence of the *cysP*, *cysU*, *cysW*, and *cysA* genes encoding an ATP-binding cassette (ABC) transporter that includes periplasmic binding proteins (Table 4). These genes may be involved in the transportation of thiosulfate or inorganic sulfate to cells, as reported in the plant-growth-promoting *Sphingomonas* sp. LK11 *Pseudomonas* sp. UW4, *Cronobacter muytjensii* JZ38, and *Pseudomonas psychrotolerans* CS51 strains [22,26,28,47]. Additionally, genes encoding the sulfite reductase (NADPH) hemoprotein beta-component (*cysI*), the sulfite reductase (NADPH) flavoprotein alpha-component (*cysJ*), cystathionine γ-lyase (*CTH*), cystathionine β-lyase (*CBS*), and thiosulfate/3-mercaptopyruvate sulfurtransferase (*TST*) (Table 4), which are known for hydrogen sulfide (H_2_S) production [47], were found. H_2_S, at low concentrations, was revealed to be an important molecule produced by PGPB with a beneficial role in modulating plant growth and development, including seed germination, root organogenesis, stomatal closure, plant maturation, and flower senescence [48]. Additionally, H_2_S improves the tolerance of plants to pathogens, osmotic stress, salt stress, heat shock, and heavy metal stresses [48], and the production of H_2_S is the other mechanism of phosphate solubilization [49]

Iron (Fe) is an essential micronutrient for plants and microorganisms and is involved in various important biological processes, such as DNA synthesis, respiration, photosynthesis, and chlorophyll biosynthesis. Fe is generally present in soils in high quantities; however, its bioavailability in neutral pH and aerobic environments, in which iron is predominantly found in the Fe^3+^ form, is extremely low and, thus, is insufficient for organisms [50]. Bacteria can overcome iron limitation by producing siderophores, low-molecular mass molecules with strong affinity for chelating Fe^3+^. A positive reaction for siderophore production was observed for the ZCR6 strain and observed as orange zones around the colonies (Table 1). In the genome of the ZCR6 strain, genes related to the biosynthesis of enterobactin (*entD*), bacterioferritin (*bfr*), and pyoverdine (*pvd*) were detected (Table 4). In addition, biosynthetic clusters of nonribosomal peptide synthetase (NRPS) engaged in pyoverdine synthesis were identified using antiSMASH analysis (Appendix A). In gram-negative bacteria, Fe-siderophore complexes are transported into the periplasm with the assistance of special receptor proteins called TonB-dependent receptors (TBDRs) [51]. The presence of the *tonB* gene encoding periplasmic protein, which provides the energy required for the translocation of siderophore–iron complexes across the bacterial outer membrane, along with the *fiu, exbB, exbD, fepA*, and *pvdF* genes encoding proteins involved in Fe-siderophore transport, was also detected in the ZCR6 genome (Table 4). Bacterial siderophores that can be efficiently used by plants have several functions in the rhizosphere [52,53,54]. Apart from increasing iron availability and thereby stimulating plant growth, siderophores can form stable complexes with other metals, such as Al, Cd, Cu, Pb, and Zn, and thus, can enhance their bioavailability. The resulting metal uptake by the plants caused by bacterial siderophores might enhance the effectiveness of the phytoremediation process [55].

#### 2.4.2. Synthesis of the Phytohormone Indole3-Acetic Acid (IAA)

The ability of rhizospheric bacteria to promote plant growth is largely due to the production of auxin phytohormones, such as indole-3-acetic acid (IAA). IAA improves plant growth by stimulating root growth through cell proliferation and tissue expansion, as well as vascular bundle and nodule formation. Moreover, IAA plays a direct role in improving plant growth under stressful conditions; for example, it may prevent the deleterious effects of heavy metals [5]. It was found that bacterial IAA can slacken the cell walls of plants and, in this way, increase the amount of root exudates. These exudates control the higher metal concentration by chelating them in the rhizosphere and preventing them from entering the plant cellular pathways [56]. It has been reported that over 80% of bacteria isolated from the rhizosphere are capable of synthesizing IAA [57,58]. This feature was also quantitatively estimated for the ZCR6 strain. The studied strain released approximately 3.6 mg IAA mL^−1^ of medium (Table 1) and was classified as a low IAA producer. The main precursor for IAA biosynthesis in bacteria is L-tryptophan. Starting with tryptophan, different pathways have been reported for the synthesis of IAA; those that are more frequently described include (1) indole-3-pyruvate, (2) tryptamine, and (3) indole-3-acetamide. Genomic analysis of the ZCR6 strain revealed the presence of tryptophan (Trp) biosynthesis genes (*trpABCDE*) along with IAA biosynthesis genes involved in the indole-3-pyruvate pathway (*aspC, aldA, aldB,* and *ipdC*) (Table 4). In this pathway, indole-3-acetaldehyde is converted from indole-3-pyruvate by indolepyruvate decarboxylase (IpdC). In addition, the ZCR6 strain carried the *iaaM* gene encoding tryptophan-2-monooxygenase (IaaM), which oxidizes tryptophan to indole-3-acetamide, but the *iaaH* gene encoding indoleacetamide hydrolase (IaaH) was not found.

#### 2.4.3. Ethylene Modulation

One of the mechanisms of plant growth promotion under various environmental stress conditions, such as the presence of organic and inorganic compounds, is the synthesis of 1-aminocyclo-propane-1-carboxylate (ACC) deaminase (ACCD), which is involved in lowering the level of ethylene. Ethylene is a phytohormone that controls the growth and maturation of plants. At low concentrations, ethylene participates in the regulation of plant growth and various metabolic processes; however, under stress conditions, plant ethylene levels increase significantly, adversely affecting the root growth and metabolism of the whole plant and leading to accelerated aging. In the genome of the ZCR6 strain, the *acdS* and *dcyD* genes, encoding ACCD and its homolog D-cysteine desulfhydrase, respectively, were detected (Table 4). Both enzymes are involved in hydrolyzing ACC, an immediate precursor of ethylene, to ammonia and α-ketobutyrate [59,60]. Consequently, the level of ACC in plant tissues is reduced, which results in a decreased level of endogenous ethylene, facilitating plant growth. In this relationship, the bacterium also benefits by utilizing the products of the reaction—nitrogen and carbon sources [61]. Biochemical characterization of the ZCR6 strain showed very high ACC deaminase activity (297 nmol α-ketobutyrate mg^−1^ h^−1^) (Table 1), which is a sufficient value to support the growth of plants in polluted soils. It was found that even a lower level of ACCD activity, approximately 20 nmol α-ketobutyrate mg^−1^ h^−1^, is sufficient to support the growth of plants in harsh environmental conditions [62]. Hong et al. [63] demonstrated a positive effect of inoculation of a rhizobacterium strain, *Gordonia* sp. S2RP-17, on the phytodegradation of diesel-contaminated soil by *Z. mays* and found a positive correlation between hydrocarbon degradation and ACC deaminase activity. Additionally, in another study, Grobelak et al. [64] demonstrated that bacteria mainly belonging to the genera *Bacillus* and *Pseudomonas* had the highest ACC deaminase activity in environments contaminated with multiple heavy metals.

Except for the abovementioned PGP mechanisms, the ZCR6 strain degraded carboxymethylcellulose (CMC) (Table 1), which indicates its ability to produce cellulase, the enzyme that plays a crucial role during the colonization of interior tissues of plants [65]. It is worth emphasizing that the ability of rhizospheric bacteria to colonize plant interiors appears to be important for establishing close interactions with plants as endophytes, especially in contaminated environments [1]. The presence of this feature indicates the potential of the strain for the colonization of plant tissues and subsequent activity as an endophyte. In this case, endophytic bacteria can protect plants against diseases (e.g., via antibiotic synthesis) and directly influence the growth of plants [66].

### 2.5. Heavy Metal Resistance

Bacteria have developed a variety of resistance mechanisms to counteract heavy metal stresses. These mechanisms include the formation and sequestration of metal, its reduction to a less toxic form, and the direct efflux of metal out of the cell [67]. It is worth noting that the beneficial effects exhibited by the metal-resistant bacteria present in the rhizosphere might have significant potential to improve phytoextraction in metal-contaminated soils [68]. In addition to the ability of the ZCR6 strain to promote plant growth, the strain was able to overcome the toxic effects of heavy metals. Genomic analysis of the ZCR6 strain revealed the presence of a number of genes responsible for the detoxification and translocation of various heavy metals, such as Cd, Co, Zn, Cu, As, Cr, and Hg (Table 5). Furthermore, the resistance of the tested strain to Cd, Zn, and Cu was confirmed in minimal inhibitory concentration tests. The MIC values for Cd, Zn, and Cu in mineral medium reached 5, 15, and 10 mM metal, respectively. Interestingly, the ZCR6 strain was able to grow in higher concentrations of cadmium than well-known *P. putida* KNP9 (2.5 mM) [67].

#### 2.5.1. Cobalt/Zinc/Cadmium Resistance

In the genome of the ZCR6 strain, the presence of a *czc* operon comprising three structural genes, *czcC, czcB,* and *czcA* (Table 5), was found. Proteins encoded by the *czc* operon constitute a branch of metal-transporting resistance-nodulation-division-type (RND-type) efflux systems, corresponding to the first layer of metal resistance in bacteria [69]. The RND family plays an important role in the heavy metal resistance of gram-negative bacteria. These pumps function as a tripartite complex composed of the inner membrane and the periplasmic and outer membrane components [70]. In the CzcCBA efflux system, CzcC is a cell wall “outer” membrane protein, CzcA is an “inner” plasma membrane transport protein, and CzcB is a membrane fusion protein functioning as a dimer. The CzcCBA efflux pump functions to transport divalent cation/proton antiporters. This efflux pump is responsible for the detoxification of Zn^2+^, Co^2+^, and Cd^2+^ [71].

#### 2.5.2. Copper/Arsenate/Chromium/Mercury Resistance

Furthermore, the ZCR6 genome also carried copper resistance genes—*copA, copB*, *cusR, cusS*, and *cueR* (Table 5). CopA and CopB are P-type Cu^+^-ATPases responsible for transporting monovalent Cu from the cytoplasm, which is coupled to ATP hydrolysis [72]. Such Cu transporters function in many gram-negative bacteria, e.g., *Pseudomonas aeruginosa* [73], *Vibrio cholerae* [74], and *Salmonella enterica* sv. *typhimurium* [75]. CusRS is a two-component, signal transduction system that is responsive to copper ions and activates the expression of the *cus*CFBA operon. In turn, CusCFBA is involved in periplasmic copper detoxification [76]; however, the genome of the ZCR6 strain lacks this system. In addition, genome analysis of the ZCR6 strain revealed the presence of the *cueR* gene encoding CueR, which regulates the expression of the P-type Cu^+^-ATPase [74].

Moreover, the presence of the *chrA* gene encoding a chromate transporter that acts as an efflux pump [77] and the *chrR* gene encoding chromate reductase that catalyzes a full reduction of Cr(VI) to Cr(III) [78] in the ZCR6 genome indicated a high degree of chromium (VI) resistance. Additionally, the arsenate resistance *arsRBC* system (Table 5) encodes ArsR, a transcriptional regulator; ArsB, an integral membrane protein that extrudes arsenite from the cell cytoplasm; and ArsC, an arsenate reductase able to transform arsenate to arsenite prior to extrusion of this metalloid [79]. ArsC enzymes use thioredoxin or glutaredoxin as electron sources; however, in the genome of the ZCR6 strain, the presence of both homologs has been reported. Additionally, the genome of ZCR6 carried the *arsH* gene-encoded ArsH protein, which was demonstrated to be an organoarsenical oxidase enzyme conferring resistance to methyl As(III) derivatives in *Pseudomonas putida* and *Sinorhizobium meliloti* [80]. It is also worth emphasizing that the genome of the ZCR6 strain carrying the *merE* gene encodes a broad mercury transporter that mediates the transport of both Hg^2+^ and CH_3_Hg(I) across bacterial membranes [81].

#### 2.5.3. Multidrug Resistance

In the genome of the ZCR6 strain, the presence of the *acrA, acrB*, and *oprM* genes encoding the AcrAB-OprM efflux system was detected (Table 5). This tripartite efflux system is composed of an inner membrane RND pump (AcrB), an outer membrane channel protein (OprM), and a periplasmic accessory protein (AcrA), and it was suggested to be responsible for multidrug resistance in gram-negative bacteria [82].

### 2.6. Exopolysaccharide (EPS) Production

The secretion of exopolysaccharides (EPS) by bacteria is a physiological process that is activated under a multitude of different environmental circumstances. The roles of these biopolymers includes basic functions such as maintaining the structural integrity of the cell envelope and enabling interactions within bacterial communities and between bacteria and eukaryotes [83]. The secretion of EPS by bacteria is also one of the most common mechanisms to protect the cell from the impact of adverse/harsh environmental conditions, such as starvation, pH, temperature, dehydration, or toxicity of heavy metals [84,85,86]. It is worth emphasizing that the genome of the ZCR6 strain carried the operons *bcs* and *alg*, involved in the biosynthesis of cellulose and alginate, respectively (Table 6). Due to the presence of various reactive groups on the polymer chains, biopolymers (including alginate and cellulose) or polymer-producing bacteria are considered to be the most attractive bioadsorbents, as they possess high selectivity and reactivity toward heavy metal ions and are recommended for the clean-up of heavy-metal-contaminated environments [87]. For example, a significant absorption capacity for Cd^2+^ or for both Cd^2+^ and Cu^2+^ was observed for *Pseudomonas* sp. strains 375 and DC-B3, respectively [88,89], indicating their potential to be used as low-cost, eco-friendly, and effective treatments for the bioremediation of heavy-metal-contaminated environments.

### 2.7. Response to Oxidative Stress

It has been observed that rhizosphere bacteria are exposed to high levels of reactive oxygen species (ROS) that are formed along the plant root surface during the reaction of redox-active plant secondary metabolites, such as phenols, quinones, flavins, and phenazines, with molecular oxygen [90]. Additionally, upon contact with bacteria, one of the plant defense reactions is the production of ROS, nitric oxide and phytoalexins; thus, rhizosphere bacteria must survive in a highly oxidative environment during the colonization of plant tissues. Moreover, the degradation of aromatic compounds leads to the generation of ROS such as H_2_O_2_, OH^−^, and O_2_. Furthermore, heavy metal ions have been reported to promote increased formation of ROS [26,91] As part of adaptation, bacteria employ various enzymes, including catalases, peroxidases, and superoxide dismutases, for defense against reactive oxygen species (ROS). In the genome of ZCR6, we found a wide variety of enzymes that help to cope with oxidative stress (Table 7). Genome analysis of the ZCR6 strain revealed genes encoding enzymes such as hydrogen peroxide catalases (*katE* and *katG*), superoxide dismutases (*sod2*), glutathione S-transferases (*GST*), gamma-glutamyltranspeptidase/glutathione hydrolase (*ggt*), glutathione peroxidase (*gpx*), and glutathione reductase (*gor*). It is also worth emphasizing that the genome of the ZCR6 strain carried the *oxyR* gene, which encodes the hydrogen peroxide sensor OxyR activating the expression of a regulon of hydrogen peroxide-inducible genes. The presence of the abovementioned genes suggests that ZCR6 possesses the potential to overcome oxidative stress, which was also confirmed by phenotypic assays.

### 2.8. Organic Compound Degradation and Biosurfactant Production

Numerous studies have shown that bacteria belonging to the genus *Pseudomonas* are known for their ability to degrade aliphatic and/or aromatic organic compounds [4,92]. For example, Muriel-Millán et al. [93] reported that the *P. aeruginosa* strain GOM1 degraded 96% of the aliphatic fraction (C12–C38) of crude oil during a 30-day incubation period, exhibiting high activity against long-chain alkanes and expressing alkane hydroxylases. In another study, Mahjoubi et al. [94] described that *Pseudomonas* sp. BUN14 could grow on and utilize various aromatic hydrocarbons, including pyrene, naphthalene, phenanthrene, carbazole, dibenzofuran, dibenzothiophene, biphenyl, pristane, fluoranthene, crude oil, octadecane, and tetradecane, as sole carbon and energy sources. It is worth noting that the effectiveness of phytoremediation supported by bacteria depends mainly on the activity of microorganisms carrying catabolic genes [95,96]. To investigate the potential of the ZCR6 strain to degrade both aliphatic and aromatic hydrocarbons, we explored its genome in search of genes related to the corresponding degradation pathways. The ZCR6 genome contains crucial genes involved in the degradation of these compounds (Table 8), which emphasizes that this strain is an excellent candidate for bioremediation.

#### 2.8.1. Degradation of Aliphatic Hydrocarbons

The genome of the ZCR6 strain carried *ladA—*a functional gene determining a novel type of alkane monooxygenase (LadA) (Table 8) that was originally detected in the genome of thermophilic *Geobacillus thermodenitrificans* NG80-2 [97]. LadA is a flavin-dependent long-chain alkane monooxygenase of the family of bacterial luciferases that utilizes a terminal oxidation pathway for the conversion of long-chain alkanes (C15 and longer, up to C36) to corresponding alcohols [98,99,100]. To the best of our knowledge, this is the first work that shows the presence of the *ladA* gene in the genome of the *Pseudomonas* genus. The ability of ZCR6 to degrade crude oil hydrocarbons was confirmed by measuring the amount of residual hydrocarbons in the culture medium during 4 weeks of cultivation. The fastest degradation rate was observed in the first week of cultivation—45.6% of the initial hydrocarbon content was degraded. In the following weeks, the rate of degradation was slower, and 76.52% of the initial hydrocarbon content was finally removed after four weeks of bacterial culture (Figure 5).

#### 2.8.2. Degradation of Aromatic Hydrocarbons

Aromatic compounds are both common growth substrates for microorganisms and prominent environmental pollutants [101]. These structurally diverse compounds are independently converted into a small number of structurally simpler common intermediates, such as catechol and protocatechuate [102]. The β-ketoadipate pathway is present in many bacterial groups, including gram-positive and gram-negative bacteria, and is considered one of the most widely distributed sets of genes for the degradation of aromatic compounds in microbes. This central pathway is composed of two ortho-cleavage branches, one for degradation of protocatechuate (*pca* genes) and the other for degradation of catechol (*cat* genes) [103]. Both protocatechuate and catechol are central metabolites derived from the oxygenation of different phenolic compounds, lignin monomers and polyaromatic hydrocarbons [104]. The final products of both pathways are acetyl-CoA and succinyl CoA [105]. It is worth noting that the protocatechuate branch of the β-ketoadipate pathway is ubiquitous in completely sequenced *Pseudomonas* genomes [28], emphasizing the importance of aromatic compound catabolism by these bacteria. Genome analysis of the ZCR6 strain revealed the presence of the *pca* cluster encoding enzymes engaged in protocatechuate catabolism but a lack of *cat* clusters. Furthermore, in the genome of the ZCR6 strain, the *hpa* genes of the homoprotocatechuate catabolic pathway were identified (Table 8). Homoprotocatechuate (3,4-dihydroxyphenylacetic acid HPC) is meta-cleaved by an HPC 2,3-dioxygenase encoded by the *hpaD* gene. The product 5-carboxymethyl-2-hydroxymuconic semialdehyde (CHMS) is then converted to 5-carboxymethyl-2-hydroxy-muconic acid (CHM) and subsequently degraded by the *hpaF, hpaG, hpaH* and *hpaI* gene products into Krebs cycle intermediates [106]. The *paa* cluster encoding enzymes involved in the phenylacetyl-CoA pathway was also present in the genome of the ZCR6 strain (Table 8). This pathway constitutes a catabolon through which many other structurally related compounds, such as phenylethylamine, phenylethanol, styrene, tropate, and n-phenylalkanoates with an even number of carbon atoms, are funneled [103]. Moreover, the ZCR6 strain contained genes for catabolism of ethylbenzene (*etbAa and etbAb*) and naphthalene (*nagZ*).

During the biodegradation of hydrocarbons, the bioavailability of these compounds is considered a key limiting factor due to their low aqueous solubility and high hydrophobicity. To enhance the bioavailability of hydrocarbons, microorganisms produce biosurfactants [107,108]. The biosurfactants produced by bacteria can also contribute to inhibiting the growth of phytopathogens, help colonize plant tissues and improve bacterial motility [109]. In this study, the ZCR6 strain produced a surface-active compound, which showed a positive result for the oil-spreading test and exhibited emulsification activities against *p*-xylene (E24 = 71.43% ± 1.43), diesel oil (E24 = 14.71% ± 0.29) and *n*-hexadecane (E24 = 25.71% ± 0.57) (Table 1). These results suggest that the ability of ZCR6 to degrade crude oil may be linked with the production of biosurfactants. For example, Cheng et al. [110] reported that *Pseudomonas aeruginosa* ZS1 isolated from petroleum sludge produced a high level of rhamnolipid that effectively emulsified crude oil, accelerating its uptake and degradation. Literature data indicate that members of the genus *Pseudomonas* possess robust metabolic machinery with the inherent ability to produce multiple and diverse secondary metabolites, including biosurfactants [111]. Genome mining by antiSMASH 5.0 resulted in the prediction of seven gene clusters associated with secondary metabolite biosynthesis (Appendix A). Many of the biosynthetic gene clusters (BGCs) are predicted to encode nonribosomal peptide synthetases (NRPSs), which produce lipopeptide (LP) and biosurfactants such as rhizomide B, viscosin, gacamide A, putisolvin, orfamide B, and anikasin. The structures of these secondary metabolites were predicted using PubChem and are presented in Appendix A. It is worth noting that LPs and rhamnolipids have been successfully used to enhance oil recovery in extreme environmental conditions [112]. In addition, bacteria use LPs for the chelation of heavy metals in a similar way to siderophores and incorporate them into their metabolism as micronutrients [113].

## 3. Materials and Methods

### 3.1. Isolation of Hydrocarbon-Degrading and Metal-Resistant Bacterial Strains

The hydrocarbon-degrading and metal-resistant strains were isolated from the rhizosphere of *Zea mays* growing in soil co-contaminated with hydrocarbons and heavy metals taken in the vicinity of the Coke Plant “Jadwiga” in Zabrze (Zabrze, Upper Silesia, Poland) using an enrichment technique. Ten grams of soil was initially suspended in 90 mL of an M9 mineral salt medium (Na_2_HPO_4_ 6 g, KH_2_PO_4_ 3 g, NaCl 0.5 g, NH_4_Cl 1 g, MgSO_4_ · 7H_2_O 0.24 g, and CaCl_2_ 0.01 g per liter of deionized water) [114] supplemented with 1% (*v*/*v*) crude oil as a carbon and energy source and 3 mM zinc as ZnSO_4_. The incubation was performed at 28 °C on a rotary shaker at 120 rpm for seven days. Continuous subculturing and shaking were carried out three times for enrichment. Next, a soil suspension was inoculated on M9 agar plates supplemented with 1% (*v*/*v*) crude oil and 3 mM ZnSO_4_, and 26 morphologically different colonies were reinoculated on M9 plates supplemented with crude oil and Zn. Isolates were then studied for their plant-growth-promoting and surface-active activities.

### 3.2. Evaluation of Plant-Growth-Promoting Activities and Surface-Active Properties

The PGP abilities of the isolates were characterized by several biochemical tests. 1-Aminocyclopropane-1-carboxylic acid (ACC) deaminase activity was assayed as described by [115] and expressed in nmol of α-ketobutyrate mg^−1^ of protein h^−1^. The protein concentration of microbial cell extracts was determined by Bradford method [116]. The cellulase activity was examined by the inoculation of the isolates on carboxymethyl cellulose agar plates as described by Pointing [117]. Siderophore secretion by the tested strain was detected by the Schwyn and Neilands method [118] using blue agar plates containing the dye Chrome azurol S (CAS). Orange zones around the colonies on the blue agar were considered positive reactions for siderophore production. Indole acetic acid (IAA) production was determined according to the modified method of Bric et al. [119] using Salkowski’s reagent. The IAA concentration in cultures was determined using a calibration curve of pure IAA (concentration range from 1 to 100 µg mL^−1^) as the standard and the production of ammonia in peptone water according to Cappuccino and Sherman [120]. The phosphate solubilizing ability of the tested strain was determined on NBRIP agar medium. Bacteria were stabbed using a sterile needle. The halo and colony diameters were measured after 14 days of incubation at 28 °C. Halo size was calculated by subtracting the colony diameter from the total diameter [121]. The production of hydrogen cyanide was tested using the method described by Lorck [122].

The surface-active properties of the strains were analyzed using the oil-spreading test and by evaluating the emulsification capacity for *n*-hexadecane, *p*-xylene, and diesel oil (by calculating the value of the emulsification index (E24)) according to the protocols described by Pacwa-Płociniczak et al. [123]. Additionally, the ability of the strains to reduce surface tension was measured using the Krűss K20 force tensiometer according to the protocol described by Ptaszek et al. [124].

All experiments were performed in triplicate and had positive (*Enterobacter intermedius* MH8b) and negative (blank, without bacterial cultivation) controls.

Based on the results of the above tests, the most promising ZCR6 strain was chosen for further analysis.

### 3.3. Biochemical Characteristics and Cellular Fatty Acid Analysis of ZCR6 Strain

The ZCR6 strain was tested for the hydrolysis of starch using starch agar flooded with iodine [125] and for the hydrolysis of gelatin using nutrient gelatin agar flooded with HgCl_2_ (15%) in HCl (20%, *v*/*v*) [126]. The activity of dehydrogenase was tested by studying the utilization of hydrogen in the presence of methylene blue solution (1% *v**/v*) [127], oxidase activity was tested on filter paper impregnated with tetramethyl-*p*-phenylenediamine [128], catalase activity was studied using broth agar flooded with H_2_O_2_ (3%, *v*/*v*) [126], urease activity was checked using nutrient broth containing urea and phenol red [129], arginine dihydrolase activity was tested using nutrient agar supplemented with L-arginine and phenol red [128], and nitrate reductase activity was tested using broth medium supplemented with KNO_3_ and nitrite test reagents [126]. Additionally, pigment formation was tested on King A and King medium B [130].

Before fatty acid extraction, bacteria were grown according to the manufacturer’s protocol on TSA agar medium at 28 °C and were reinoculated three times on fresh TSA medium to stabilize the fatty acid composition. Initially, fatty acids were released and saponified by the addition 1mL of reagent 1 (45 g NaOH, 150 mL methanol, 150 mL water) to the sample and incubation at 100 °C for 30 min with occasional vortexing. Next, free fatty acids were methylated by adding reagent 2 (325 mL 6 N HCl, methanol 275 mL) solution (2 mL) and incubated at 80 °C for 10 min. Phase separation was then carried out by the addition 1 mL of reagent 3 (1:1 mixture of n-hexane and methyl-tert butyl ether) with vigorous mixing by vortex for 30 s. After sufficient phase separation, the top organic phase was transferred to a new glass tube and washed with 3 mL of NaOH solution (10.8 g NaOH, 900 mL water). The top organic phase was again transferred to a new GC vial and analyzed according to the standard protocol of MIDI (Sherlock Microbial Identification System, version 6.0) and identified using the TSBA 6.21 database of the Microbial Identification System as described in [131].

### 3.4. DNA Extraction, Whole Genome Sequencing, and Assembly

A single colony of the ZCR6 strain taken from an M9 agar plate supplemented with crude oil and Zn was reinoculated into 30 mL of liquid M9 medium with 1% (vol/vol) crude oil and 3 mM Zn and incubated at 28 °C on a rotary shaker at 120 rpm for 48 h. Genomic DNA was extracted using the GeneMatrix Bacterial and Yeast Genomic Purification Kit according to the manufacturer’s instructions (EURx^®^, Gdańsk, Poland). Genomic libraries were constructed using the NEBNext^®^ DNA Library Prep Master Mix Set for Illumina^®^ and sequenced using the Illumina MiSeq platform with 2 × 250 bp paired-end reads (Illumina, San Diego, SD, USA). The primer sequences were trimmed out of the reads using Cutadapt 1.16 (phred score cutoff: 20; maximum trimming error rate: 0.1; minimum required adapter overlap: 3 bp). Reads were assembled using SPAdes 3.11.1 (to generate contigs, default parameters were used; error correction and Kmer sizes were set to auto). Genome sequences were deposited in the GenBank database under BioProject accession number PRJNA681194 under accession number JADWSX000000000.

### 3.5. Genome Functional Annotation

Functional annotation of genes was performed using a multitude of tools and databases, such as the NCBI Prokaryotic Genome Annotation Pipeline (PGAP) (www.ncbi.nlm.nih.gov/genome/annotation_prok/, accessed on 1 December 2020), the eggNOG 5.0 orthology prediction pipeline (http:/eggnogdb.embl.de, accessed on 15 May 2020) [132], and the PATRIC tool v.3.6.9 (https://www.patricbrc.org, accessed on 15 May 2020) [133]. For annotation of gene function, genes were compared to the KEGG database (Functional Kyoto Encyclopedia of Genes and Genomes database) (http://www.genome.jp/kegg/, accessed on 15 May 2020) [134]. Functional and pathway analyses were also performed using the BlastKOALA web tool of KEGG (https://www.kegg.jp/blastkoala/, accessed on 15 May 2020) [134]. A subsequent annotation was performed using Rapid Annotations using a Subsystems Technology tool kit (RASTtk) found in PATRIC tool v.3.6.9. Functional analysis by Cluster of Orthologous Genes (COGs) was performed with the use of WebMGA (http://weizhong-lab.ucsd.edu/webMGA/server/cog/, accessed on 15 May 2021) [135]. Identification of gene clusters responsible for biosynthesis of secondary metabolites was performed using antiSMASH 5.0 (https://antismash.secondarymetabolites.org, accessed on 15 May 2021) [136]. Prophage sequences were identified and annotated using PHAST [137]. The circular genome was generated with the PATRIC tool v.3.6.9. For genome functional annotation, all the default parameters were used for the analysis.

### 3.6. Phylogenetic Analysis

To check the taxonomic placement of the ZCR6 strain, the 16S rRNA gene was amplified with the universal bacterial primers 8F (5′ AGTTTGATCATCGCTCAG 3′) and 1492R (5′ GGTTACCTTGTTACGACTT 3′), targeting a fragment size of 1484 bp [107]. Next, the obtained sequence with a length of 1402 bp was compared with the EZBioCloud database. The phylogenetic analysis was performed based on the longest common fragment of the 16S rRNA gene sequences (1387 bp) selected from ClustalW alignment of ZCR6 strains and 13 type strains of other *Pseudomonas* species. Sequences of 16S rRNA fragments of type strains were obtained from GenBank. The analysis was performed using Mega X software with the maximum likelihood method and the Tamura-3-parameter model, assuming that a certain fraction of sites are evolutionarily invariable (+I) and 1000 bootstrap replicates [138,139]. The similarity of 16S rRNA sequences between tested and type strains was calculated using the average nucleotide identity (ANI) calculator (https://www.ezbiocloud.net/tools/ani, accessed on 12 September 2021) [140].

Detailed identification of the ZCR6 strain was obtained using multilocus sequence analysis (MLSA) of the partial sequences of 16S rRNA and two housekeeping genes (*gyrB and rpoD*) [20,138]. After alignment using ClustalW, the concatenated sequences (2563 bp for 16S rRNA-*gyrB*-*rpoD* genes) were compared with concatenated sequences of type strains obtained from GenBank (accession numbers of sequences are given in Appendix A). Phylogenetic analysis of the ZCR6 strain was performed using Mega X software with the maximum likelihood method with 1000 bootstrap replicates under the Tamura–Nei model using the discrete gamma distribution (+G) [138]. The similarity of concatenated gene sequences between tested and type strains was calculated using the ANI calculator. Additionally, using the TrueBac ID tool (truebacid.com, accessed on 16 November 2020), the isolate was identified based on the sequence of the entire genome [141]. The target genes were extracted from the entire genome using Geneious Prime software.

A tetra-correlation search (TCS), a feature of JSpeciesWS (http://jspecies.ribohost.com/jspeciesws, accessed on 15 May 2020) [142], allowed comparisons of the genome sequenced of the ZCR6 strain against the genomes from the reference database GenomesDB, providing a list of the most similar genomes based on their resulting tetranucleotide signature correlation index.

### 3.7. Hydrocarbon Degradation Analysis

The ability of the ZCR6 strain to degrade crude oil was studied in 0.1 LB (Luria Bertani) broth. Replicate flasks containing 500 mL of medium supplemented with crude oil at a 1% concentration (*v*/*v*) were prepared. The flasks were inoculated with 5 mL of one-week-old liquid culture of the ZCR6 strain. Non-inoculated flasks were prepared as controls. The cultures were grown aerobically at 28 °C for 4 weeks with constant shaking (120 rpm). To determine the amount of hydrocarbon degraded, the initial concentration of hydrocarbons and the residual hydrocarbon concentration after 4 weeks of incubation were determined.

The mineral oil content was estimated according to PN-EN ISO 9377-2. The quantitative determination of mineral oil (hydrocarbon) content after extraction was estimated by gas chromatography. The 0.1 LB medium with crude oil was extracted by mechanical shaking with an acetone/*n*-heptane mixture for 1 h at 120 rpm. The organic layer was separated and washed twice with water to remove acetone. The remaining *n*-heptane extract was dried using anhydrous Na_2_SO_4_. Polar compounds were removed by adsorption on Florisil^®^/Na_2_SO_4_ columns. An aliquot of the purified extract was analyzed by GC-FID. The total peak area in the range C_10_-C_40_ was measured, and the amount of hydrocarbons in the sample was quantified against an external standard.

The total petroleum hydrocarbon (TPH) content was analyzed using a gas chromatograph (Agilent 7820A) equipped with a flame ionization detector and an Rxi-5ms capillary column (25 m × 0.2 mm ID × 0.33 µm); the injection volume was 1 µL, and hydrogen was used as the carrier gas (gas flow 2.1 mL minute^−1^). The operation program was started with injector and detector temperatures of 300 °C. The oven temperature was initially programmed at 60 °C, held for 10 min, increased to 320 °C at 30 °C minute^−1^, and then held for 10 min.

### 3.8. Minimal Inhibitory Concentration (MIC) Determination

The level of resistance to heavy metals of the ZCR6 strain was assessed by determination of the minimal inhibitory concentration (MIC) of Zn^2+^, Cu^2+^, Cd^2+^, and Ni^2+^ ions as described by [143]. The MIC values were determined in triplicate in Tris minimal salt medium with glucose (Tris-MSM) (Tris base 1 g, glucose 5 g, NaCl 0.5 g, KCl 0.5 g, MgSO_4_ · 7H_2_O 0.2 g, (NH_4_)_2_SO_4_ 0.5 g, CaCl_2_ 0.01, FeSO_4_ · 7H_2_O 0.001 g per liter of deionized water with pH range of 7.0 ± 0.2.) [144]. The Tris-MSM medium was supplemented with increasing concentrations of the metals (0.5–20 mM). The inoculum was prepared by culturing the ZCR6 strain in LB medium at 28 °C on a rotary shaker at 120 rpm for 24 h. After incubation, the culture was centrifuged at 5000 rpm for 20 min, washed twice with sterile sodium chloride (0.9%), and resuspended in Tris-MSM medium. Media amended with different concentrations of heavy metals were inoculated with bacterial inoculum to obtain an optical density (OD) of 0.1 at 600 nm. As for the controls, medium with inoculum but without metal and medium amended with metals but without inoculum were prepared. Cultures were incubated at 28 °C on a rotary shaker at 120 rpm, and the OD was measured after 72 h. The MIC was defined as the lowest metal concentration at which bacterial growth was not observed.

## 4. Conclusions

In this study, we confirmed that the *P. qingdaonensis* ZCR6 strain isolated from the rhizosphere of maize exhibits potential for the bioremediation of sites co-contaminated with hydrocarbons and heavy metals using this plant. However, the potential contamination of plant biomass should be considered when conducting the phytoremediation experiments using crop plants, e.g., maize. On the other hand, in most cases, the seeds of plants are protected, and heavy metals are not accumulated in such structures; thus, their application is possible. Nevertheless, phytoremediation should be conducted under controlled conditions, and plant biomass cannot be used as food for animals and humans.

The examined strain shows the potential to promote plant growth through a wide range of mechanisms, such as the production of IAA, ACC deaminase, siderophores, ammonia and solubilization of phosphate. It is worth noting that these mechanisms can also affect the effectiveness of the bioremediation process. Additionally, sequencing of the ZCR6 strain genome provided important information about features that may be crucial for bioremediation. Notably, genetic analyses were linked with the outcomes of previously performed assays, expanding the understanding of genome–phenotype interplays. We identified the genes involved in the metabolism and degradation of xenobiotics. Furthermore, genome sequencing of ZCR6 opened up a number of opportunities to study the mechanisms involved in the production of secondary metabolites via PGP, including biosurfactants and metal resistance. We believe that our research provided substantial interesting data about a new *P*. *qingdaonensis* species. Complete characteristics of the ZCR6 strain are beneficial for fully understanding the mechanisms involved in the bioremediation of co-contaminated sites.

## Figures and Tables

**Figure 1 ijms-23-00214-f001:**
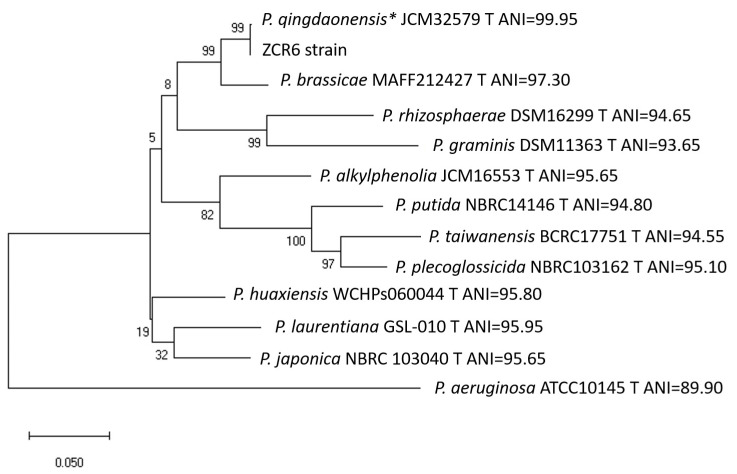
Maximum-likelihood-based phylogeny resulting from the concatenated 16S rRNA, *gyrB*, and *rpoD* sequences, representing the relative position of the ZCR6 strain among other *Pseudomonas*. All positions containing gaps or missing data were eliminated. The final dataset contained 2563 positions. Accession numbers of sequences used in this study are listed in Appendix A. Bootstrap values are represented at the branching points. The bar represents 0.05 substitutions per site. Each strain is given an ANI value relative to the sequences belonging to the ZCR6 strain; * strain JJ3^T^ with the highest ANI value.

**Figure 2 ijms-23-00214-f002:**
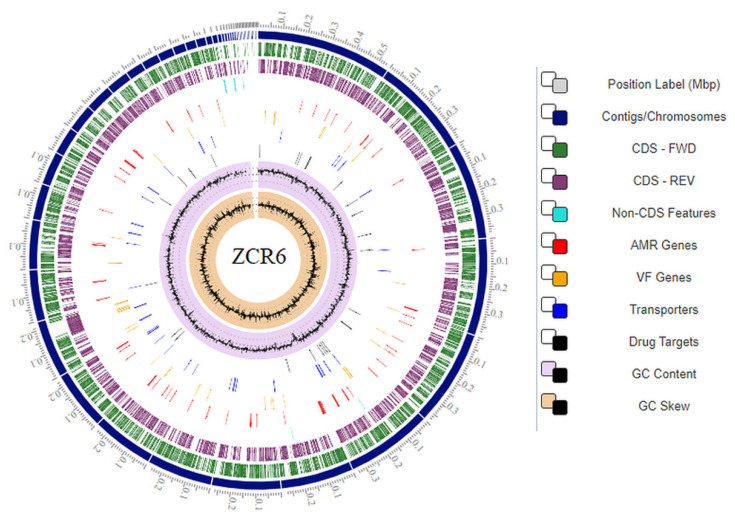
Circular plot representing the genome annotations of the ZCR6 strain. Colors depict the different classification types of gene clusters along the sequenced genome. The description of each circle is represented from the outermost circle to the innermost circle. Tick marks represent the predicted CDS on the positive and negative strands. AMR—antimicrobial resistance genes; VF—virulence factor genes.

**Figure 3 ijms-23-00214-f003:**
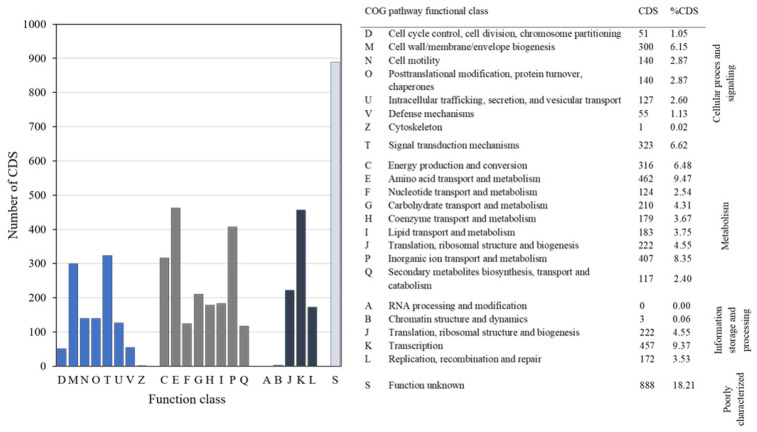
COG classification of predicted genes in the ZCR6 strain. Colored bars indicate the CDS assigned to each COG category.

**Figure 4 ijms-23-00214-f004:**
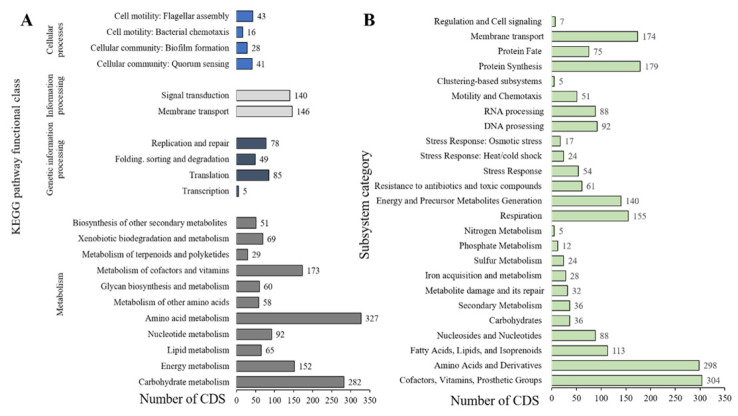
KEGG pathway classification of predicted genes in the ZCR6 strain using KEGG BlastKoala (**A**). Protein-coding sequences grouped into the functional subsystems in the ZCR6 genome using RASTtk (**B**).

**Figure 5 ijms-23-00214-f005:**
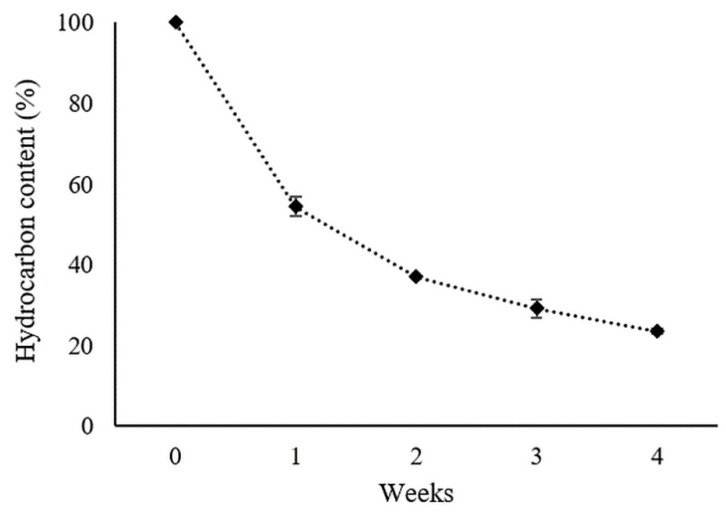
Degradation of hydrocarbons by the ZCR6 strain (mean ± SD, n D 3).

**Table 1 ijms-23-00214-t001:** The activity of the plant-growth-promoting mechanisms and surface-active abilities of the ZCR6 strain.

PGP Mechanisms						Plant Colonisation	Surface-Active Properties				
ACCD ^a^	IAA ^b^	Ca_3_(PO_4_)_2_ ^c^	Siderophore ^d^	HCN ^e^	NH_3_ ^f^	CMC ^g^	Surface Tension ^h^	Oil-Spreading ^i^	Emulsification Index ^j^		
									H ^k^	X ^l^	D ^m^
296.73 ± 1.27	3.59 ± 0.11	6.5 ± 0.53	+	-	+	+	68.1 ± 0.55	4.0 ± 0.2	25.71 ± 0.57	71.43 ± 1.43	14.71 ± 0.29

^a^ ACCD activity (nmol a-ketobutyrate mg^−1^ h^−1^); ^b^ (IAA mL^−1^ of medium); ^c^ the halo diameter (mm) calculated by subtracting the colony diameter from the total halo size; ^d–g^ + positive; − negative; ^h^ (mN m^−1^); ^i^ (mm); ^j^ (%); ^k^ H—*n*-hexadecane; ^l^ X—*p*-xylene; ^m^ D—diesel oil.

**Table 2 ijms-23-00214-t002:** Fatty acid composition (%) of the ZCR6 strain.

	Percentage (%)	
	24 h	72 h
C16:0	30.95	22.52
C16:1ω7c and/or C16:1ω6c	26.35	11.42
C18:1ω7c and/or C18:1ω6c	15.44	11.68
C17:0 cyclo	10.22	23.07
C12:0 2OH	4.63	8.74
C12:0 3OH	4.11	7.57
C10:0 3OH	3.34	4.35
C12:0	2.22	4.07
C 19:0 cyclo ω8c	<1	5.14
Others	<2	<2

**Table 3 ijms-23-00214-t003:** Genome features of *P. qingdaonensis* ZCR6.

Feature	Value
Assembly information	
Genome size (bp)	5,507,067
Contigs	52
Conting N50 (bp)	292,270
Conting L50	7
G + C content (%)	64.5
Annotation information	
Genes (total)	5055
CDSs (total)	4979
Genes (coding)	4943
CDSs (with protein)	4943
Pseudogenes	36
RNA genes	76
rRNA	3, 3, 2 (5S, 16S, 23S)
tRNA	64
ncRNA	4
Prophage regions	2
Genes assigned to KEGG pathways	2945
Genes assigned to KEGG Orthology (KO)	2209
Genes assigned to COGs	4877
Genes connected to RASTtk	2319
BioProject ID	PRJNA681194
BioSample ID	SAMN16935590
GeneBank accession number	JADWSX000000000

**Table 4 ijms-23-00214-t004:** Products of the genes involved in the most important plant growth promotion traits present in the genome of the ZCR6 strain.

Gene	Accession Number	KO Number	Gene Product	Activity
*iscS*	MBG8561789.1	K04487	Cysteine desulfurase	Nitrogen fixation
*iscU*	MBG8560681.1	K04488	Nitrogen fixation protein nifu	
*gcd*	MBG8562075.1	K00117	Quinoprotein glucose dehydrogenase (EC:1.1.5.2)	Phosphate solubilization
*pqqA*	MBG8558041.1	K06135	Pyrroloquinoline quinone biosynthesis protein A	
*pqqB*	MBG8558042.1	K06136	Pyrroloquinoline quinone biosynthesis protein B	
*pqqC*	MBG8558043.1	K06137	Pyrroloquinoline-quinone synthase (EC:1.3.3.11)	
*pqqD*	MBG8558044.1	K06138	Pyrroloquinoline quinone biosynthesis protein D	
*pqqE*	MBG8558045.1	K06139	Pqqa peptide cyclase (EC:1.21.98.4)	
*ppa*	MBG8560327.1	K01507	Inorganic pyrophosphatase (EC:3.6.1.1)	
*ppx-gppA*	MBG8558276.1	K01524	Exopolyphosphatase/guanosine-5′-triphosphate,3′-diphosphate pyrophosphatase (EC:3.6.1.11 3.6.1.40)	
*pstB*	MBG8558408.1	K02036	Phosphate transport system ATP-binding protein (EC:7.3.2.1)	
*pstA*	MBG8558409.1	K02038	Phosphate transport system permease protein	
*pstC*	MBG8558410.1	K02037	Phosphate transport system permease protein	
*pstS*	MBG8558411.1	K02040	Phosphate transport system substrate-binding protein	
*pit*	MBG8560129.1	K16322	Low-affinity inorganic phosphate transporter	
*phoB*	MBG8558402.1	K07657	Two-component system. Ompr family. Phosphate regulon response regulator phob	
*phoR*	MBG8558403.1	K07636	Two-component system. Ompr family. Phosphate regulon sensor histidine kinase phor (EC:2.7.13.3)	
*phoU*	MBG8558407.1	K02039	Phosphate transport system protein	
*ssuA*	MBG8561374.1	K15553	Sulfonate transport system substrate-binding protein	Sulfur metabolism
*ssuB*	MBG8560999.1	K15555	Sulfonate transport system ATP-binding protein (EC:7.6.2.14)	
*ssuC*	MBG8561372.1	K15554	Sulfonate transport system permease protein	
*ssuD*	MBG8561373.1	K04091	Alkanesulfonate monooxygenase (EC:1.14.14.5)	
*ssuE*	MBG8561375.1	K00299	FMN reductase (EC:1.5.1.38)	
*cysP*	MBG8561508.1	K02048	Sulfate/thiosulfate transport system substrate-binding protein	
*cysU*	MBG8561415.1	K02046	Sulfate/thiosulfate transport system permease protein	
*cysW*	MBG8561414.1	K02047	Sulfate/thiosulfate transport system permease protein	
*cysA*	MBG8561413.1	K02045	Sulfate/thiosulfate transport system ATP-binding protein (EC:7.3.2.3)	
*cysI*	MBG8558903.1	K00381	Sulfite reductase (NADPH) hemoprotein beta-component (EC:1.8.1.2)	
*cysJ*	MBG8560698.1	K00380	Sulfite reductase (NADPH) flavoprotein alpha-component (EC:1.8.1.2)	
*CTH*	MBG8560530.1	K01758	Cystathionine γ-lyase	
*CBS*	MBG8560531.1	K01697	Cystathionine β-lyase	
*TST*	MBG8560836.1	K01011	Thiosulfate/3-mercaptopyruvate sulfurtransferase	
*bfr*	MBG8560256.1	K03594	Bacterioferritin (EC:1.16.3.1)	Siderophore biosynthesis
*entD*	MBG8562760.1	K02362	Enterobactin synthetase component D (EC:6.3.2.14 2.7.8.-)	
*fiu*	MBG8560699.1	K16090	Catecholate siderophore receptor	Siderophore uptake
*tonB*	MBG8561215.1	K03832	Periplasmic protein tonb	
*exbB*	MBG8561408.1	K03561	Biopolymer transport protein exbb	
*exbD*	MBG8558382.1	K03559	Biopolymer transport protein exbd	
*fepA*	MBG8559634.1	K19611	Ferric enterobactin receptor	
*pvdF*	MBG8561616.1	K06160	Putative pyoverdin transport system ATP-binding/permease protein	
*trpC*	MBG8557998.1	K01609	Indole-3-glycerol phosphate synthase (EC:4.1.1.48)	Synthesis of the phytohormone:
*trpD*	MBG8557999.1	K00766	Anthranilate phosphoribosyltransferase (EC:2.4.2.18)	indole acetic acid (IAA)
*trpG*	MBG8558000.1	K01658	Anthranilate synthase component II (EC:4.1.3.27)	
*trpE*	MBG8558001.1	K01657	Anthranilate synthase component I (EC:4.1.3.27)	
*trpF*	MBG8560957.1	K01817	Phosphoribosylanthranilate isomerase (EC:5.3.1.24)	
*trpA*	MBG8562725.1	K01695	Tryptophan synthase alpha chain (EC:4.2.1.20)	
*trpB*	MBG8562726.1	K01696	Tryptophan synthase beta chain (EC:4.2.1.20)	
*ipdC*	MBG8559993.1	K04103	Indolepyruvate decarboxylase (EC:4.1.1.74)	
*aspC*	MBG8558823.1	K00813	Aspartate aminotransferase (EC:2.6.1.1)	
*aldA*	MBG8559624.1	K07248	Lactaldehyde dehydrogenase/glycolaldehyde dehydrogenase (EC:1.2.1.22 1.2.1.21)	
*aldB*	MBG8560334.1	K00138	Aldehyde dehydrogenase (EC:1.2.1.-)	
*iaaM*	MBG8558038.1	K00466	Amine oxidase	
*acdS*	MBG8559878.1	K01505	1-aminocyclopropane-1-carboxylate deaminase (EC:3.5.99.7)	Ethylene modulation
*dcyD*	MBG8561384.1	K05396	D-cysteine desulfhydrase (EC:4.4.1.15)	

**Table 5 ijms-23-00214-t005:** Genes potentially involved in metal resistance in the ZCR6 genome.

Gene	Accession Number	KO Numbers	Gene Product	Metal Resistance
*czcA*	MBG8562084.1	K15726	Cobalt-zinc-cadmium resistance protein czca	Cobalt/zinc/cadmium resistance
*czcB*	MBG8562085.1	K15727	Membrane fusion protein. Cobalt-zinc-cadmium efflux system	
*czcC*	MBG8562086.1	K15725	Outer membrane protein. Cobalt-zinc-cadmium efflux system	
*copB*	MBG8561457.1	K01533	P-type Cu^2+^ transporter (EC:7.2.2.9)	Cooper resistance
*copA*	MBG8562586.1	K17686	P-type Cu^+^ transporter (EC:7.2.2.8)	
*cusR*	MBG8559773.1	K07665	Two-component system. Ompr family. Copper resistance phosphate regulon response regulator cusr	
*cusS*	MBG8559774.1	K07644	Two-component system. Ompr family. Heavy metal sensor histidine kinase cuss (EC:2.7.13.3)	
*cueR*	MBG8562587.1	K19591	Merr family transcriptional regulator. Copper efflux regulator	
*arsR*	MBG8562772.1	K03892	Arsr family transcriptional regulator. Arsenate/arsenite/antimonite-responsive transcriptional repressor	Arsenate resistance
*arsB*	MBG8559938.1	K03893	Arsenical pump membrane protein	
*arsC*	MBG8559939.1	K03741	Arsenate reductase (thioredoxin) (EC:1.20.4.4)	
*arsC*	MBG8562772.1	K00537	Arsenate reductase (glutaredoxin) (EC:1.20.4.1)	
*arsH*	MBG8559940.1	K11811	Arsenical resistance protein arsh	
*chrA*	MBG8559121.1	K07240	Chromate transporter	Chromium resistance
*chrR*	MBG8561757.1	K19784	Chromate reductase. Nad(p)h dehydrogenase (quinone)	
*oprM*	MBG8560149.1	K18139	Outer membrane protein. Efflux system	Multidrug resistance
*acrB*	MBG8560150.1	K18138	Multidrug efflux pump	
*acrA*	MBG8560151.1	K03585	Membrane fusion protein. Multidrug efflux system	
*acrR*	MBG8560152.1	K03577	Tetr/acrr family transcriptional regulator. Acrab operon repressor	
*merE*	MBG8560549.1	K00549	5-methyltetrahydropteroyltriglutamate--homocysteine methyltransferase (EC:2.1.1.14)	Mercury resistance

**Table 6 ijms-23-00214-t006:** Genes involved in exopolysaccharides production.

Gene	Accession Number	KO Numbers	Gene Product	Activity
*bcsA*	MBG8561468.1	K00694	Cellulose synthase (UDP-forming) (EC:2.4.1.12)	Exopolysaccharides biosynthesis
*bcsB*	MBG8561469.1	K20541	Cellulose synthase operon protein B	
*bcsZ*	MBG8561470.1	K20542	Endoglucanase (EC:3.2.1.4)	
*bcsC*	MBG8561471.1	K20543	Cellulose synthase operon protein C	
*algF*	MBG8561473.1	K19296	Alginate O-acetyltransferase complex protein algf	
*algI*	MBG8561474.1	K19294	Alginate O-acetyltransferase complex protein algi	
*algJ*	MBG8561475.1	K19295	Alginate O-acetyltransferase complex protein algj	
*algB*	MBG8562662.1	K11384	Two-component system. Ntrc family. Response regulator algb	
*algA*	MBG8559808.1	K16011	Mannose-1-phosphate guanylyltransferase/mannose-6-phosphate isomerase (EC:2.7.7.13 5.3.1.8)	
*algL*	MBG8560007.1	K01729	Poly(beta-D-mannuronate) lyase (EC:4.2.2.3)	
*algX*	MBG8560008.1	K19293	Alginate biosynthesis protein algx	
*algG*	MBG8560009.1	K01795	Mannuronan 5-epimerase (EC:5.1.3.37)	
*algE*	MBG8560010.1	K16081	Alginate production protein	
*algK*	MBG8560011.1	K19292	Alginate biosynthesis protein algk	
*alg44*	MBG8560012.1	K19291	Mannuronan synthase (EC:2.4.1.33)	
*alg8*	MBG8560013.1	K19290	Mannuronan synthase (EC:2.4.1.33)	
*algD*	MBG8560014.1	K00066	GDP-mannose 6-dehydrogenase (EC:1.1.1.132)	
*algH*	MBG8561216.1	K07735	Putative transcriptional regulator	
*algR*	MBG8558298.1	K08083	Two-component system. Lyttr family. Response regulator algr	

**Table 7 ijms-23-00214-t007:** Genes potentially involved in response to oxidative stress in the ZCR6 genome.

Gene	Accession Number	KO Numbers	Gene Product	Activity
*katE*	MBG8562679.1	K03781	Catalase (EC:1.11.1.6)	Oxidative stress response
*katG*	MBG8559566.1	K03782	Catalase-peroxidase (EC:1.11.1.21)	
*sod2*	MBG8560099.1	K04564	SOD2; superoxide dismutase. Fe-Mn family (EC:1.15.1.1)	
*ggt*	MBG8558271.1	K00681	Gamma-glutamyltranspeptidase/glutathione hydrolase (EC:2.3.2.2 3.4.19.13)	
*gst*	MBG8558300.1	K00799	Glutathione S-transferase (EC:2.5.1.18)	
*gpx*	MBG8560587.1	K00432	Glutathione peroxidase (EC:1.11.1.9)	
*gsr*	MBG8561128.1	K00383	GSR; glutathione reductase (NADPH) (EC:1.8.1.7)	
*oxyR*	MBG8558139.1	K04761	Lysr family transcriptional regulator. Hydrogen peroxide-inducible gene activator	

**Table 8 ijms-23-00214-t008:** Genes involved in organic compounds’ degradation.

Gene	Accession Number	KO Numbers	Gene Product	Activity
*ladA*	MBG8560652.1	K20938	long-chain alkane monooxygenase (EC:1.14.14.28)	Aliphatic compounds’ degradation
*pcaG*	MBG8562545.1	K00448	Protocatechuate 3.4-dioxygenase. Alpha subunit (EC:1.13.11.3)	Aromatic compounds’ degradation
*pcaH*	MBG8562546.1	K00449	Protocatechuate 3.4-dioxygenase. Beta subunit (EC:1.13.11.3)	
*pcaQ*	MBG8560766.1	K02623	Lysr family transcriptional regulator. Pca operon transcriptional activator	
*pcaC*	MBG8560139.1	K01607	4-carboxymuconolactone decarboxylase (EC:4.1.1.44)	
*pcaD*	MBG8560138.1	K01055	3-oxoadipate enol-lactonase (EC:3.1.1.24)	
*pcaR*	MBG8560131.1	K02624	Iclr family transcriptional regulator. Pca regulon regulatory protein	
*pcaK*	MBG8560132.1	K08195	MFS transporter. AAHS family. 4-hydroxybenzoate transporter	
*pcaF*	MBG8560135.1	K07823	3-oxoadipyl-coa thiolase (EC:2.3.1.174)	
*pcaT*	MBG8560136.1	K02625	MFS transporter. MHS family. Dicarboxylic acid transporter pcat	
*pcaB*	MBG8560137.1	K01857	3-carboxy-cis.cis-muconate cycloisomerase (EC:5.5.1.2)	
*hpaF*	MBG8558716.1	K01826	5-carboxymethyl-2-hydroxymuconate isomerase (EC:5.3.3.10)	
*hpaI*	MBG8558890.1	K02510	4-hydroxy-2-oxoheptanedioate aldolase (EC:4.1.2.52)	
*hpaH*	MBG8558891.1	K02509	2-oxo-hept-3-ene-1.7-dioate hydratase (EC:4.2.1.-)	
*hpaF*	MBG8558893.1	K01826	5-carboxymethyl-2-hydroxymuconate isomerase (EC:5.3.3.10)	
*hpaD*	MBG8558894.1	K00455	3.4-dihydroxyphenylacetate 2.3-dioxygenase (EC:1.13.11.15)	
*hpaE*	MBG8558895.1	K00151	5-carboxymethyl-2-hydroxymuconic-semialdehyde dehydrogenase (EC:1.2.1.60)	
*hpaG*	MBG8558896.1	K05921	5-oxopent-3-ene-1.2.5-tricarboxylate decarboxylase/2-hydroxyhepta-2.4-diene-1.7-dioate isomerase (EC:4.1.1.68 5.3.3.-)	
*hpaA*	MBG8558898.1	K02508	Arac family transcriptional regulator. 4-hydroxyphenylacetate 3-monooxygenase operon regulatory protein	
*paaF*	MBG8559324.1	K01692	Enoyl-coa hydratase (EC:4.2.1.17)	
*paaX*	MBG8559346.1	K02616	Phenylacetic acid degradation operon negative regulatory protein	
*paaY*	MBG8559347.1	K02617	Phenylacetic acid degradation protein	
*paaG*	MBG8559349.1	K15866	2-(1.2-epoxy-1.2-dihydrophenyl)acetyl-coa isomerase (EC:5.3.3.18)	
*paaH*	MBG8559350.1	K00074	3-hydroxybutyryl-coa dehydrogenase (EC:1.1.1.157)	
*paaI*	MBG8559351.1	K02614	Acyl-coa thioesterase (EC:3.1.2.-)	
*paaK*	MBG8559353.1	K01912	Phenylacetate-coa ligase (EC:6.2.1.30)	
*paaA*	MBG8559354.1	K02609	Ring-1.2-phenylacetyl-coa epoxidase subunit paaa (EC:1.14.13.149)	
*paaB*	MBG8559355.1	K02610	Ring-1.2-phenylacetyl-coa epoxidase subunit paab	
*paaC*	MBG8559356.1	K02611	Ring-1.2-phenylacetyl-coa epoxidase subunit paac (EC:1.14.13.149)	
*paaD*	MBG8559357.1	K02612	Ring-1.2-phenylacetyl-coa epoxidase subunit paad	
*paaE*	MBG8559358.1	K02613	Ring-1.2-phenylacetyl-coa epoxidase subunit paae	
*paaZ*	MBG8559362.1	K02618	Oxepin-coa hydrolase/3-oxo-5.6-dehydrosuberyl-coa semialdehyde dehydrogenase (EC:3.3.2.12 1.2.1.91)	
*etbAa;*	MBG8559252.1	K14748	Ethylbenzene dioxygenase subunit alpha (EC:1.14.12.-)	
*etbAb;*	MBG8559253.1	K14749	Ethylbenzene dioxygenase subunit beta (EC:1.14.12.-)	

## Data Availability

Genome sequences were deposited in the GenBank database under BioProject accession number PRJNA681194 under accession number JADWSX000000000 (https://www.ncbi.nlm.nih.gov/nuccore/JADWSX000000000, accessed on 1 December 2020). 16S rRNA gene sequence was deposited in the GenBank database under accession number OK597188.1 (https://www.ncbi.nlm.nih.gov/nuccore/OK597188, accessed on 1 December 2020).

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
