# Peer review of "Analysis of the Genome of the Heavy Metal Resistant and Hydrocarbon-Degrading Rhizospheric Pseudomonas qingdaonensis ZCR6 Strain and Assessment of Its Plant-Growth-Promoting Traits"

_ijms, 2021, doi:10.3390/ijms23010214_

Round 1
Reviewer 1 Report
The manuscript “Analysis of the genome and biochemical characterization reveal plant growth promotion, hydrocarbon degradation and heavy metal resistance traits of the new rhizospheric pseudomonas qingdaonensis ZCR6 strain” by Chlebek D et al., presents isolation of the strain Pseudomonas qingdaonensis ZCR6 from the rhizosphere of maize grown in soil co-contaminated with hydrocarbons and heavy metals. Further analysis of the stain revealed plant growth promotion potential, heavy metal resistance, potential to overcome oxidative stress, and its involvement in organic compound degradation and biosurfactant production. Overall, the study is good and further exploration of such bioinoculants would help understand the mechanisms involved in the bioremediation of contaminated soils.
- However, it is not very clear from the methods how were authors able to isolate strain ZCR6 from 26 different strains? Include a detailed isolation method
- Line 183, it was written 0.1 LB medium, please clarify
- I suggest authors to write in detail the methodology of how FAMEs were extracted
- For the read trimming and filtering, mention the parameters that were used for discarding the low-quality reads
- For genome functional annotation, mention if all the default parameters were used for the analysis. If parameters were changed for any of the analysis, please include
- Change the font size, line 259, 260, 536, 611.
- Under results, general features of the genome, include Longest and small contig sizes N50 & L50 values, and if pseudo genes were identified
- I suggest authors to provide 300 dpi images for figure 2, 3, & 4
- Though may not be absolutely necessary for the paper but for better controls and comparisons, usage of the strain that is closed related from the phylogenetic analysis would have been very helpful to make the investigation more thorough.
Author Response
Dear Rewiever,
We would like to thank you for very constructive and detailed comments that helped us to improve our manuscript. Answers to your remarks are in the attachment.
Sincerely yours
Magdalena Pacwa-Płociniczak

Reviewer 2 Report
Dear author
This manuscript reported that a genome data and some characteristics of a novel strain of Psedumonas. Pseudomonas qingdaonensis is anovel species and its characteristics are not well-known. Therefore this report has a worth to be published in scientific journal. I recommend authors to demonstrate the plant-growth promoting activity with a bioassay using any plants. And the Pseudomonas qingdaonensis is not validated as the species by LPNS. Therefore, authors should explain it in the manuscript. Analysis of the genome is well written.
Sincerely yours
Author Response
Dear Rewiever,
We would like to thank you for the comments. Answers to your remarks are in the attachment.
Sincerely yours
Magdalena Pacwa-Płociniczak

Reviewer 3 Report
26 strains were isolated from the corn rhizosphere in contaminated land. One of the strains, ZCR6, was sequenced and investigated further through a set of in vitro bioassays and biochemical analysis.
- Lab assays miss necessary controls. At the very basic level, these assays require Blank and one other strain in addition to ZCR6.
- The manuscript in general lacks important experiments. I was keen to see some data that the ZCR6 promotes the growth in corn because the in vitro assays do not reflect the field performance. ZCR6 may produce a certain growth promoter but it may as well secret some growth inhibitors. We would not know the overall performance unless we test on the plant.
- The authors make many speculative conclusions based on the presence of certain genes in ZCR6. They completely disregard that a gene should be expressed to do the job potentially. In this context, transcriptional regulations, expression networks, protein post modifications, etc, were overlooked. With this in mind, a gene has many alleles functioning at different levels and as it is common knowledge such allele diversity would lead to natural variations. Some alleles contain nonsense mutation and its subsequent biological consequence is unlikely to be captured by HTP sequencing employed in this manuscript.
- Authors claim that ZCR6 has “great potential for the bioremediation of sites co-contaminated with hydrocarbons and heavy metals.” We are talking about enhancing crop growth on contaminated land, which may or may not cause a high concentration of heavy metals in a crop that is going to enter the food supply chain, either directly or indirectly (animal feeding). A paper like this must be mindful of such undesirable effects and quantify heavy metals content in the plant materials before publishing and promoting this strain. On the other hand, if the use of ZCR6 would not lead to a high concentration of heavy metals in the plant (i.e. removal of the heavy metals from soil), then all the discussions and statements regarding the bioremediation would be irrelevant to this manuscript.
- Throughout the manuscript, there are quite a few statements that are hard to understand, either need more elaborations or should be reworded. For example:
- L271: “nucleotide identity was observed for P. qingdaonensis (99.95%), P. brassicae (97.30%) and P. laurentiana (95.95%) (Fig. 1).” What does this mean, please elaborate? With such a high identity why was P. laurentiana clustered so far away from ZCR6?
- L535: Could authors explain how cellulases “plays a crucial role during colonization of plants”? This enzyme digest cellulose to monosaccharides i.e. aims to disintegrate the plant cell wall! The reference was cited for this claim is not relevant and does not make such a statement at all !!
- The first line of the result starts with “…. In this study, bacterial strain ZCR6 was isolated…” The authors know much about the ZCR6 but the readers don’t! and they would like to know the process of screening and selections that led to the isolation of ZCR6.
- There are too many tables in the manuscript, some contain only one row of data! for instance, Lines 287-290 convey the exact same information as in table1!
- L304-308: Do you mean the difference between the two analyses is due to the incubation period? If so, does your conclusion mean ZCR6 is the same as JJ3T strain? Did you try such analysis after 3 days of incubation to test your claim?
- The “P. qingdaonensis strain JJ3T” has been mentioned many times but why this strain is not present on the phylogeny tree.
- The paper is kind of disorganised and unnecessarily long, with many repeated claims. The title reads “ 3.1. Isolation and identification of the isolate” but there is nothing about isolation and identification of ZCR6, just some basic phylogenetic analysis. This section misses describing what the other 26 strains are. Why did you choose ZCR6? Such incongruity can be found in many places in the paper.
- Legend of fig5: requires more explanation and details. What are the error bars, how many times the experiment was repeated?
- L256-258: does not belong to the results section
- L 274-277: do not belong to the result section
- L 287, L311 and many other occasions: As it is widely accepted, the full name of Pseudomonas qingdaonensis only need to be mentioned once, afterwards it is P. qingdaonensis
- L311-313: Up to this point you have not shown a single data to prove this!!
Round 2
Reviewer 3 Report
Appreciate the authors' efforts, they addressed all the concerns adequately.
Line 363: May need to add “fatty acid methyl esters” before FAME.
Regarding point 4, “great potential for the bioremediation”. It is up to the authors but some cautiousness may be needed while discussing maize for bioremediation and certainly adding this part of your response “seeds of plants are protected, and heavy metals are not accumulated in such structures” to the conclusion would be helpful. There has been some good research on using perennial biomass crops for bioremediation at the University of Silesia, those scientists always voiced their concern about growing food crops on polluted lands.
Good luck with your publication.
Author Response
Dear Reviewer,
Thank you for appreciating our work and your kind words. According to your suggestions, we added the whole name of FAME analysis (line 363) and we added information about cautiousness that is needed while discussing maize for bioremediation (lines 799-805).